# Immune Checkpoint Inhibitors and Vaccination: Assessing Safety, Efficacy, and Synergistic Potential

**DOI:** 10.3390/vaccines12111270

**Published:** 2024-11-11

**Authors:** Jacob New, Luke Shenton, Radia Ksayer, Justin Wang, Karam Zakharia, Laura J. Nicholson, Amitabh C. Pandey

**Affiliations:** 1Department of Medicine, Scripps Health, La Jolla, CA 92121, USA; new.jacob@scrippshealth.org (J.N.); shenton.luke@scrippshealth.org (L.S.); wang.justin@scrippshealth.org (J.W.); nicholson.laura@scrippshealth.org (L.J.N.); 2Scripps Research Translational Institute, La Jolla, CA 92037, USA; 3Department of Medicine, Tulane University, New Orleans, LA 70112, USA; rksayer@tulane.edu (R.K.); kzakharia@tulane.edu (K.Z.); 4Section of Cardiology, Department of Medicine, Tulane University, New Orleans, LA 70112, USA; 5Southeast Louisiana Veterans Health Care System, New Orleans, LA 70112, USA

**Keywords:** immune checkpoint inhibitors (ICIs), cancer vaccination, immunotherapy, mRNA vaccines, infectious disease vaccination

## Abstract

Although immune checkpoint inhibitors (ICIs) have become predominant therapies for cancer, the safety and efficacy of combining ICIs with vaccinations remain areas of needed investigation. As ICIs gain broader clinical application, the relevance of current vaccination guidelines for cancer patients—largely developed in the context of cytotoxic therapies—becomes increasingly uncertain. Although data support the safety of combining inactivated influenza and mRNA SARS-CoV-2 vaccines with ICI therapy, comprehensive data on other infectious disease vaccines remain scarce. Notably, the combination of ICIs with infectious disease vaccines does not appear to exacerbate immune-related adverse events, despite the heightened cytokine activity observed. However, the efficacy of vaccines administered alongside ICIs in preventing infectious diseases remains poorly supported by robust evidence. Preliminary findings suggest a potential survival benefit in cancer patients receiving ICI therapy alongside influenza or SARS-CoV-2 vaccination, though the quality of evidence is currently low. Moreover, the synergistic potential of combining therapeutic cancer vaccines, particularly mRNA-based vaccines, with ICIs indicates promise but with a paucity of phase III data to confirm efficacy. This review critically examines the safety and efficacy of combining ICIs with both infectious disease vaccines and therapeutic cancer vaccines. While vaccination appears safe in patients undergoing ICI therapy, the impact on infectious disease prevention and cancer treatment outcomes warrants further rigorous investigation.

## 1. Introduction

Immune checkpoint inhibitors (ICIs) have significantly advanced cancer therapy by leveraging the immune system’s capacity to target and eliminate cancer cells. However, the precise mechanisms by which ICIs modulate the immune response, especially in the context of their interactions with vaccines, remain only partially understood [1,2]. As ICI therapy becomes more widespread, understanding the safety and efficacy of administering vaccinations concurrently with ICIs has become increasingly important.

The existing vaccination guidelines, primarily based on experience with cytotoxic therapies, do not fully address the complexities introduced by checkpoint inhibition, which fundamentally alters T-cell activity [1]. This necessitates an evaluation of vaccination strategies within this therapeutic landscape, especially as the combination of ICIs and therapeutic cancer-antigen-targeted vaccines gains increasing clinical interest.

Emerging evidence suggests that while ICIs enhance the immune response against tumors, they may also influence the immune system’s response to vaccines in ways that are not yet fully elucidated. The immunomodulatory effects of ICIs, which involve the amplification of T-cell-mediated responses, could theoretically enhance the efficacy of concurrent vaccinations. However, these interactions also raise concerns about the potential for exacerbated immune-related adverse events (irAEs), as well as the impact on vaccine efficacy in terms of both infectious disease prevention and cancer treatment outcomes.

Cancer patients are particularly vulnerable to infections, facing a mortality risk from infectious diseases that is three times higher than that of the general population [3]. This elevated risk stems from chronic inflammation associated with malignancy, compromised immune cell function, and the immunosuppressive effects of cytotoxic treatments [4,5]. Compounding these challenges, cancer patients are often under-immunized [6,7]. The widespread utilization of ICIs, with their potential for long-lasting responses and a unique adverse effect profile, underscores the need to carefully assess how vaccination strategies should be adapted for this patient population.

## 2. Checkpoint Inhibitors: Prevalence, Mechanism, and Immunization Safety Considerations

In 2019, approximately 43% of cancer patients were deemed eligible for immunotherapy, a number that has risen due to the broadening of approved indications for ICIs [8]. This increasing prevalence emphasizes the critical need to understand how ICIs’ mechanism of action may interact with vaccine-induced immune responses. ICIs, including agents targeting CTLA-4 and PD-1 axes, are designed to disrupt inhibitory pathways that typically restrain the immune system to avoid autoimmunity. By blocking these checkpoints, ICIs lift these inhibitory signals, thereby enabling T-cells to mount a more robust attack against cancer cells.

Nevertheless, the heightened immune activation facilitated by ICIs can precipitate irAEs, which are observed in an estimated 44% of patients undergoing therapy [9]. Among these, more common adverse effects—experienced by over 10% of patients—include fatigue, rash, pruritus, and diarrhea. Less frequent irAEs, occurring in 1–10% of cases, involve more severe conditions such as colitis, hypothyroidism, pneumonitis, hepatitis, and arthralgia. Although rare, affecting less than 1% of patients, severe irAEs like severe pneumonitis, neurological complications, and myocarditis can pose significant risks. Given this context, it is critical to evaluate immune-provoking vaccination strategies in combination with ICI therapy for the potential risk of exacerbating irAEs.

Concerns were initially raised when a single-center reported an elevated risk of grade 3 and 4 irAEs following inactivated influenza vaccination, compared to historical controls [10]. There was preexisting preclinical evidence of a potential mechanism where vaccination in conjunction with ICIs led to an increased cytokine response [11]. However, subsequent evidence from larger studies suggests that vaccination is generally safe for patients on ICIs and does not significantly heighten the risk of irAEs [12]. Nonetheless, the initial concerns underscore the importance of thoroughly assessing the safety and efficacy of various vaccine types when administered alongside ICI therapy.

Cardiovascular complications linked to ICI therapy, such as myocarditis, pericardial diseases, Takotsubo syndrome, arrhythmias, and vasculitis, are often under-recognized, partly due to their non-specific symptoms like fatigue, weakness, and shortness of breath [13,14]. These symptoms frequently mirror those of vaccine-related acute adverse effects, complicating the timely diagnosis and reporting of these conditions. This overlap necessitates heightened awareness when monitoring patients undergoing vaccination while on ICIs.

As the clinical use of ICIs continues to expand, it becomes increasingly crucial to consider not only the potential risks of exacerbating irAEs but also the broader implications for patient safety, particularly in relation to vaccination strategies.

## 3. Infectious Disease Vaccinations and ICI Therapy

Infectious diseases pose a significant threat to cancer patients, particularly those who are immunocompromised due to their underlying malignancy or the therapies they receive. The risk of severe illness from infections such as influenza and COVID-19 is markedly higher in this population, necessitating careful consideration of vaccination strategies to mitigate these risks [15,16]. Furthermore, the humoral response to vaccination is not as strong among cancer patients as compared to patients without cancer [17]. A thorough examination of the risks and benefits of vaccination in cancer patients is essential, with a focus on the different types of vaccines available, and their associated safety profiles. Given the contraindications associated with live vaccines in immunocompromised individuals, attention is given to inactivated and mRNA vaccines, which are more commonly recommended in this vulnerable population.

Vaccines can be broadly categorized into live, inactivated, and mRNA vaccines, each with distinct characteristics and implications for cancer patients receiving ICI therapy. Live attenuated vaccines contain weakened forms of the pathogen that can replicate within the host, typically providing robust and long-lasting immunity [18]. However, in immunocompromised individuals, including many cancer patients, these vaccines pose a significant risk as they can potentially cause disease due to the weakened state of the patients’ immune system. Examples of live vaccines include the MMR (measles, mumps, and rubella), oral typhoid, yellow fever, rotavirus, nasal influenza, and varicella vaccines.

In contrast, inactivated vaccines contain pathogens that have been killed or inactivated, meaning they cannot replicate but still stimulate an immune response. These vaccines are generally considered safer for immunocompromised patients, though the immune response they elicit may be less robust compared to live vaccines. Vaccines made of purified antigens, bacterial components, and genetically engineered recombinant antigens further refine this approach by isolating specific parts of the pathogen, such as proteins or polysaccharides, to stimulate immunity without introducing the whole organism.

The mRNA vaccines, a newer class exemplified by the COVID-19 vaccines, work by instructing cells to produce a protein that triggers an immune response. These vaccines do not contain live virus and do not integrate into the host’s genome, making them a promising option for immunocompromised patients. In this review, we refer to mRNA vaccines both in the context of SARS-CoV-2 vaccines and more broadly to explore their potential interaction with ICI therapy beyond COVID-19. While the clinical experience with SARS-CoV-2 mRNA vaccines has been substantial, generalizing insights from these vaccines to other mRNA-based platforms warrants caution. Therefore, where appropriate, we specify whether the findings relate specifically to SARS-CoV-2 mRNA vaccines or reflect broader mRNA vaccine technology. This distinction ensures a balanced discussion and avoids bias in interpreting the safety, efficacy, and immune responses associated with mRNA vaccines.

Guidelines from the Infectious Disease Society of America (IDSA) and the National Comprehensive Cancer Network (NCCN) underscore the importance of carefully selecting vaccines for cancer patients. According to the IDSA and NCCN clinical practice guidelines, live vaccines should be avoided in immunocompromised hosts due to the potential risks [19]. NCCN additionally highlights that monoclonal antibody therapy is a risk factor for infection [20]. Given these guidelines, the focus for patients receiving ICI therapy should be on inactivated and mRNA vaccines, which offer a safer profile while still providing protection against infectious diseases (Figure 1).

### 3.1. Inactivated Vaccines and ICI Therapy

Among inactivated vaccines, the most robust data are available for the influenza and COVID-19 vaccines, both of which have been studied in the context of ICI therapy. The influenza vaccine has been the subject of numerous studies evaluating its safety in cancer patients receiving ICIs. However, data on other inactivated vaccines, such as those for pneumococcus, hepatitis B, and tetanus, remain limited. Only one single-institution study provides evidence on the safety of all inactivated vaccines [21], which underscores the need for further investigation into the safety and effectiveness of these vaccines in patients on ICI therapy.

Additionally, there are limited data regarding the inactivated COVID-19 vaccine, which is not FDA approved, used in combination with immunotherapy agents, a large number of which were also not FDA approved [22,23]. It is important to note that there was decreased seroconversion observed in ICI-treated patients to inactivated SARS-CoV-2 vaccination [23]. Yet, with the geographic differences in ICI therapies assessed, it remains difficult to understand the generalizability of these findings.

### 3.2. Influenza Vaccination in the Context of ICI Therapy

Substantial evidence supports the safety and potential benefits of inactivated influenza vaccination in cancer patients. Analysis of data from multiple studies, encompassing over a thousand patients, has consistently shown that influenza vaccination in combination with ICI therapy is well tolerated [12,24]. IrAEs following vaccination remain low, with only a small proportion of patients experiencing grade 3–4 toxicities, indicating that the vaccine is safe in the context of ICI therapy.

Beyond its favorable safety profile, influenza vaccination appears to induce a strong immune response in patients treated with ICIs. Notably, seroprotection rates in ICI-treated patients have been found to surpass those seen in patients receiving traditional cytotoxic therapies, suggesting that ICIs may enhance the immune response to inactivated influenza vaccines rather than impede it [25]. Yet, there is a need to more fully understand the cytokine profile associated with this humoral response.

The efficacy of the influenza vaccine in preventing influenza infection among ICI-treated patients remains an area of needed investigation. While the robust humoral response observed in these patients is promising, studies have not yet demonstrated a significant reduction in the incidence of influenza infections following vaccination [26,27]. The limited number of studies is primarily due to the low overall reported incidence of flu-like symptoms among ICI-treated patients. As such, further research is needed to clarify the extent to which the immunogenicity of the influenza vaccine translates into real-world protection against influenza in patients receiving ICI therapy.

### 3.3. mRNA Vaccines and ICI Therapy

The advent of mRNA vaccines has introduced a novel approach to immunization particularly relevant to patients treated with ICI therapy. These vaccines, which work by delivering genetic instructions for the production of specific viral proteins, have demonstrated remarkable efficacy and safety profiles in the general population. Due to the extensive data available for SARS-CoV-2 mRNA vaccines, many insights presented in this section reference these vaccines. However, emerging studies are beginning to evaluate other mRNA platforms for both infectious and cancer-directed vaccinations. While SARS-CoV-2 data provide a useful framework, the translation of these findings to other mRNA vaccines offers a potential additional therapy. SARS-CoV-2 mRNA vaccines result in an initial cytokine response of IFN- γ, IL-2 and IL-4, alongside an antibody and T-cell response, which result in durable clinical protection against infection, though for a limited amount of time [28,29]. For cancer patients, mRNA vaccines offer a promising alternative due to their ability to induce robust immune responses without the risk associated with live pathogens. As with inactivated vaccines, the integration of mRNA vaccines into the treatment regimens of ICI-treated patients raises important questions regarding safety, efficacy, and the potential for immune-related adverse events.

### 3.4. SARS-CoV-2 mRNA Vaccination and ICI Therapy

The COVID-19 pandemic has underscored the critical need for effective vaccines, particularly for cancer patients, who are at an elevated risk of severe disease due to their immunocompromised status. The antibody response to SARS-CoV-2 mRNA vaccination in cancer patients varies depending on the type of underlying malignancy and the specific anticancer therapy received [30,31,32]. Given these variations, it is crucial to assess not only the antibody response but also the cytokine profile and clinical efficacy outcomes of mRNA vaccination when administered in conjunction with ICI therapy.

The interaction between mRNA vaccines and ICI therapy produces a notable cytokine response. Cytokine release syndrome (CRS) is a rare complication of ICIs [33]. Yet, the combination of these agents results in a CRS-like response observed alongside administration of the SARS-CoV-2 mRNA vaccine with ICI therapy. A case study of a patient with colorectal cancer on anti-PD-1 monotherapy initially highlighted the risk of developing CRS shortly after SARS-CoV-2 mRNA vaccination [34]. This patient exhibited elevated levels of several cytokines, notably IFN-γ, IL-2R, IL-16, and IL-18. The temporal association between vaccination and CRS suggests a potential trigger effect as the ICI therapy may have amplified the immune response leading to CRS. This was further evaluated in a series of 35 patients who received SARS-CoV-2 mRNA vaccination while receiving ICI therapy. Again, CRS-like responses occurred in these patients, with the predominant cytokines being IL-6, CXCL8, IL-2, CCL2, and soluble IL-1 receptor alpha [35]. Although the cytokine profile in these patients was detected serologically, there was no clinically observed CRS response greater than grade II in both short- and long-term follow-up. The cytokine profile of mRNA vaccination administered during ICI therapy in these small studies suggests a different profile than what occurs when the mRNA vaccine is administered to non-ICI-treated patients, yet there remains a need for further studies to compare the differing cytokine profiles.

It is also important to observe that these increased cytokine responses related to the SARS-CoV-2 mRNA vaccines may be unique to this particular formulation. Both the spike protein encoded or the lipid nanoparticle may be the instigator of the enhanced cytokine response observed in patients [36].

With an enhanced cytokine response, there is concern that concurrent mRNA vaccination and ICI therapy may result in a heightened risk of irAEs. To date, there has not been a study that directly compares the rate of irAEs in ICI-treated patients among mRNA-vaccinated and unvaccinated patients [32]. In six studies that report irAE frequency in ICI-treated patients who received mRNA vaccination, the observed rate of irAE is no more than 23.6%, which would correspond to a rate similar or slightly better than the population of patients treated with ICI therapy [32]. Yet, the overall quality of evidence to estimate the rate of irAE after mRNA immunization remains low.

Myocarditis is a notable irAE and potential complication of mRNA vaccination that warrants closer examination as new mRNA vaccines continue to be developed. Although both ICI-induced and mRNA vaccine-induced myocarditis may share a similar mechanism involving the overactivation of the immune response, the presenting characteristics of myocarditis differ between the two [37]. A common symptom observed in mRNA-vaccine-induced myocarditis was fever, yet this occurred in less than 50% of ICI-induced myocarditis. Similarly, dyspnea occurred in 67% of ICI-induced myocarditis, but less than 15% of mRNA-vaccine-induced myocarditis. Although the left ventricular ejection fraction was similar among ICI- and mRNA-induced myocarditis, ICI-induced myocarditis had a lower left ventricular global longitudinal strain and lower atrial conduit strain, suggesting different mechanisms for these adverse effects. No study has examined the pathophysiology of myocarditis in the combination of ICI and mRNA vaccination, yet given the differing clinical syndromes of the disease, it is crucial to investigate whether the combination of these therapies may exacerbate the risk or alter the presentation of myocarditis. Understanding the nuanced differences in the mechanisms and clinical manifestations of myocarditis related to ICI therapy and mRNA vaccination will be essential as these treatments become more widely used in cancer patients. Future research should focus on elucidating the underlying pathophysiology when these therapies are combined, to ensure that effective monitoring and management strategies are in place to mitigate potential risks. Additionally, although myocarditis has been observed with both ICI therapy and SARS-CoV-2 mRNA vaccination, the mechanism may be driven by the spike protein encoded by the SARS-CoV-2 vaccines. Therefore, myocarditis may not be a generalizable adverse effect across all mRNA vaccines. Careful monitoring and further evaluation of myocarditis risks with other mRNA vaccines are essential, particularly in ICI-treated patients.

Antibody response to the SARS-CoV-2 mRNA vaccine appears similar among ICI-treated patients and patients without cancer [32]. Additionally, the common combination of chemotherapy and immunotherapy did not result in significantly less antibody response to mRNA vaccination compared to immunotherapy alone [31]. Notably, despite observing an enhanced cytokine response, there does not appear to be an increased titer of antibodies. The antibody response among ICI-treated patients remains durable for months following immunization.

Despite observing a similar antibody response and an enhanced cytokine response, it remains unclear if these findings translate to clinical protection from SARS-CoV-2. Despite receiving four vaccinations, patients undergoing ICI therapy remain at a higher risk of clinically detectable COVID-19 compared to the healthy population [38]. This finding was distinct among cytotoxic therapies evaluated, showing a comparable effect size to the post-vaccination infection risk observed with CD20-directed antibody therapy or calcineurin inhibitor therapy [38]. Thus, it remains essential to understand if surrogate markers of response, such as cytokine profile or antibody titer, translate to clinical protection from infection when mRNA vaccinations and ICI therapy are combined.

### 3.5. Enhanced Overall Survival with the Combination of Infection-Targeted Vaccines in ICI Therapy

Several studies have intriguingly observed that combining infectious disease-targeted vaccination with ICI therapy results in improved overall survival (Table 1). This was first observed in patients with metastatic solid malignancies receiving PD-1- or PD-L1-targeted ICI therapy, where inactivated influenza vaccination administered either 60 days before or during therapy was linked to improved progression-free survival and overall survival. These benefits remained significant even after adjusting for age, Charlson Comorbidity Index, and treatment line [39]. A prospective study designed to evaluate the efficacy of an influenza vaccine educational intervention in ICI-treated patients also demonstrated improved progression-free and overall survival in those administered the influenza vaccine during ICI therapy [40].

Notably, this survival advantage is not limited to the influenza vaccine. Patients receiving SARS-CoV-2 mRNA vaccination while on ICI therapy also demonstrated improved overall survival [35]. These findings suggest that infection-targeted vaccines may play a synergistic role with ICIs, contributing to better clinical outcomes.

While these findings are promising, several limitations must be considered. First, most of the studies demonstrating improved overall survival with the combination of infectious disease-targeted vaccination and ICI therapy are retrospective or observational in nature. This study design inherently limits the ability to establish causality and introduces potential biases, such as patient selection bias and unmeasured confounding factors.

Additionally, the survival benefits observed might be influenced by other variables not fully accounted for, such as variations in disease burden, the timing of vaccination relative to ICI administration, or differences in supportive care practices across study populations. The observed effects might also be partially due to the healthier subset of patients who are more likely to receive vaccinations and survive long enough to benefit from ICI therapy.

Furthermore, while some studies have adjusted for key variables like age, comorbidities, and treatment line, the lack of randomized controlled trials specifically designed to evaluate survival outcomes in this context makes it difficult to draw definitive conclusions. As a result, the generalizability of these findings across different cancer types, stages, and patient populations remains uncertain. Further prospective, randomized studies are needed to validate these observations and better understand the potential synergistic effects and optimal timing of combining ICIs with vaccines. Importantly, a mechanistic understanding of the factors driving the observed survival benefits is necessary. Identifying and studying biological correlates associated with these outcomes should be strongly encouraged, as they could offer insights into the underlying mechanisms and guide future therapeutic strategies.

## 4. Cancer-Directed Vaccines and ICI Therapy

Therapeutic cancer vaccines represent a promising and innovative direction in cancer immunotherapy, offering a personalized approach that could potentially enhance both efficacy and tolerability [41]. These vaccines are designed by identifying tumor-associated antigens and delivering them through various platforms, including tumor lysates, tumor DNA, viral vectors, mRNA, and peptides [42]. Once delivered, these antigens are processed by antigen-presenting cells, thereby inducing a targeted immune response against the patient’s tumor. A particularly noteworthy subset of these antigens, known as neoantigens, arises from the unique genetic mutations found in cancer cells. Given the remarkable success of ICI therapy, there is growing interest in combining therapeutic cancer vaccines with ICI treatment to explore potential synergistic effects that could further improve clinical outcomes.

There is significant interest in the development of RNA vaccines with ICI therapy to improve cancer outcomes [43]. This was bolstered by the encouraging safety profile observed of mRNA vaccines during the COVID-19 pandemic. The need for a combination of mRNA vaccines with an immunostimulatory agent, such as ICI, was promoted when a phase I trial of patients with metastatic gastrointestinal cancer treated with tumor antigen mRNA resulted in no objective responses [44]. In a phase I trial, the safety of an RNA vaccine in combination with ICI therapy for melanoma was observed, alongside durable objective responses [45]. A phase IIb study demonstrated an mRNA-based neoantigen therapy alongside ICI therapy in stage IIIb-IV melanoma patients availed a tolerable safety profile and prolonged recurrence-free survival [46]. This observation has resulted in the study vaccine being the first mRNA vaccine to move into a phase III trial. Beyond melanoma, mRNA particles have been combined with ICI therapy in pancreatic cancer, and in a phase I study, the administration was found to be safe. Additionally, a longer median recurrence-free survival was observed in patients who developed vaccine-expanded T-cells [47].

Different methods of mRNA administration have also been trialed alongside ICI therapy. A phase I/II study of a priming adenoviral vaccine followed by lipid nanoparticle encapsulated self-amplifying mRNA resulted in durable T-cell response and an observation of increased overall survival in microsatellite-stable colorectal cancer [48]. Yet, when evaluating this vector system with mRNA encoding 20 shared neoantigens in a phase I/II study of patients with advanced/metastatic solid tumors in combination with ICI, there was no response observed [49]. This discrepancy underscores the critical need to understand the biological mechanisms of immunostimulation when vaccines are administered alongside ICIs. Additionally, a phase II study of tumor antigen mRNA-loaded dendritic cells and ICI therapy resulted in durable tumor responses in patients with pretreated advanced melanoma [50]. Yet, there are no reported phase III studies detailing the response to combination tumor antigen vaccines and ICI therapy.

While there is growing interest in mRNA vaccines for cancer due to their versatility and the safety profile observed during the COVID-19 pandemic, other vaccine strategies, including peptide-based and cell-based vaccines, have demonstrated significant synergy with ICIs. For example, long-peptide vaccines combined with ICIs have shown promising outcomes in recurrent HPV-driven malignancies. A phase II trial reported a 33% overall response rate with a long-peptide HPV-16 vaccine plus ICI therapy, highlighting the potential for non-mRNA platforms to complement ICIs [51].

Dendritic cells serve as platforms to present tumor-associated antigens, thereby enhancing T-cell activation. For instance, a phase I trial administering dendritic cells loaded with tumor antigens alongside ICIs in patients with stage IV melanoma resulted in stable disease for four participants [52]. This emphasizes the importance of carefully defining vaccine platforms, as cell-based vaccines do not represent a simple combination of dendritic cells and peptides, but rather a targeted antigen-presentation strategy. Certainly, a variety of preclinical studies are evaluating different vaccine platforms, as reviewed recently [53]. As these platforms move into clinical studies alongside ICI therapy, it will be essential to observe the potential adverse effect profile, with a particular focus on cytokine response. Furthermore, it is essential to recognize that other combination strategies beyond cancer-directed vaccines are being investigated with ICIs, as reviewed recently [54].

While mRNA vaccines are a focal point due to the rapid clinical advancements made during the COVID-19 pandemic, it is essential to explore other vaccine types. The variability in clinical outcomes based on cancer type, vaccine platform, and delivery method highlights the need for more comprehensive research. Phase III trials evaluating the long-term benefits of combining ICIs with different cancer vaccine platforms remain limited, and further research is crucial to confirm the efficacy and safety of these approaches.

Lastly, it is also important to consider that the cytokine response observed with infection-targeted vaccines may offer survival benefits. Non-cancer antigens, such as those in influenza and SARS-CoV-2 vaccines, could stimulate immune pathways that enhance outcomes when combined with ICIs. As such, therapeutic vaccine trials must carefully choose comparator arms to account for these non-cancer antigen effects, which could confound the evaluation of tumor antigen-specific efficacy.

## 5. Conclusions

Despite the evidence of safety associated with influenza and SARS-CoV-2 vaccines in patients receiving ICI therapy, there remain significant underexplored areas that warrant further investigation. While much of the current research has focused on these two vaccines, other vaccines, such as those targeting pneumococcus, zoster, hepatitis B, tetanus, and HPV, have not been studied as extensively in the context of ICI therapy. Understanding the potential interactions between these less-studied vaccines and immunotherapy is crucial, particularly as more vaccines are developed and recommended for this vulnerable population, and considering the lack of evidence of clinical protection. Furthermore, given the recognized cardiac toxicities associated with both ICI therapy and mRNA vaccination, a concerted effort is needed to establish monitoring practices of cardiac function in patients receiving these therapies concurrently.

Additionally, the potential overall survival benefit observed with the combination of infection-targeted vaccines and ICI therapy is an intriguing finding that deserves further exploration. While initial studies suggest a synergistic effect that enhances survival outcomes, more robust clinical trials are necessary to confirm these benefits and elucidate the underlying mechanisms driving this potential survival advantage.

Although cancer patients have been historically less likely to be vaccinated, it is encouraging that cancer survivors have an increased rate of vaccination [55]. Given the durable responses observed in patients treated with ICI, broader efforts should be taken to promote education regarding vaccine safety and potential benefits.

Overall, vaccines are generally safe for patients on immunotherapy, with no significant association between vaccinations and the occurrence of irAEs. However, the outcomes related to vaccines against diseases other than influenza and COVID-19 remain underexplored. Additionally, there is a need to evaluate more fully whether adequate protection from post-vaccination infection occurs if the vaccine is administered while receiving ICI therapy. Future research should prioritize investigating additional vaccines, along with the potential survival benefits observed with infection-targeted vaccinations, to ensure that cancer patients receive the most effective and comprehensive care possible.

## Figures and Tables

**Figure 1 vaccines-12-01270-f001:**
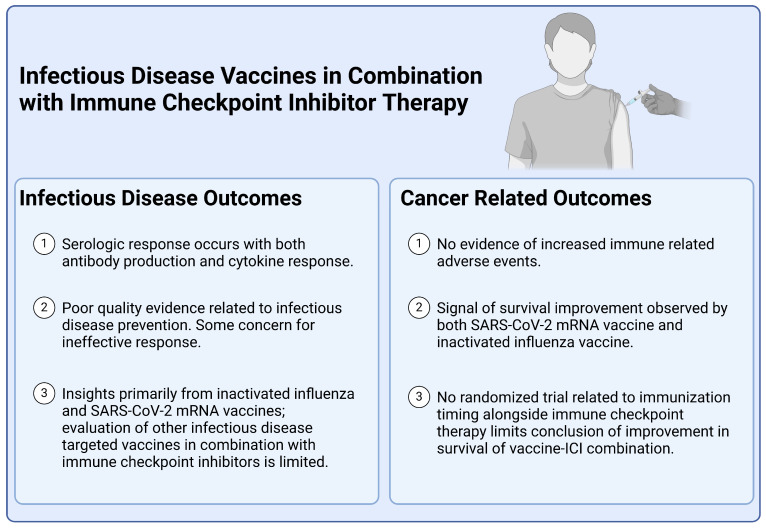
Effects of Infectious Disease Vaccination Combined with Immune Checkpoint Inhibitor Therapy on Infectious Disease and Cancer Outcomes.

**Table 1 vaccines-12-01270-t001:** Oncologic outcomes following combination of infection-targeted vaccines and ICI therapy.

Trial Design	Intervention	Outcomes	Toxicities	Reference
Multicenter retrospective cohort study	Influenza vaccination during treatment with immune checkpoint inhibitor, or 60 days prior to treatment initiation. Mixed population of malignancy types. Intervention (vaccinated), n = 67; non-vaccinated control, n = 236.	6-month PFS HR = 0.63, 95% CI: 0.41–0.98 after adjusting for age, gender, CCI, PS, CNS metastasis, and line of therapy. 6-month OS HR = 0.53, 95% CI: 0.30–0.93 after adjusting for age, gender, CCI, PS, CNS metastasis, and line of therapy.	No difference in rate of irAE between groups.	[37]
Multicenter prospective observational study	Influenza vaccination during treatment with immune checkpoint inhibitor. Mixed population of malignancy types. Intervention (vaccinated), n = 581; non-vaccinated control, n = 607.	PFS HR = 0.85, 95% CI: 0.72–1.01 after propensity score matching for age, sex, smoking habits, primary tumor site, comorbidity, and PS. OS HR = 0.75, 95% CI: 0.62–0.92 after propensity score matching for age, sex, smoking habits, primary tumor site, comorbidity, and PS.	No difference in vaccine-related adverse events. irAE rate not reported.	[38]
Single-center retrospective and prospective cohort study	COVID-19 mRNA vaccination during treatment with immune checkpoint inhibitor. Mixed population of malignancy types. Intervention (vaccinated), n = 64; non-vaccinated control, n = 26.	OS HR = 0.21, 95% CI: 0.07–0.69.	No difference in irAE among matched cohorts.	[14]

Abbreviations used: PFS, progression-free survival; HR, hazard ratio; CCI, Charlson Comorbidity Index; PS, performance status; CNS, central nervous system; OS, overall survival; irAE, immune-related adverse event.

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
