# Peer review of "Immune Checkpoint Inhibitors and Vaccination: Assessing Safety, Efficacy, and Synergistic Potential"

_vaccines, 2024, doi:10.3390/vaccines12111270_

Round 1
Reviewer 1 Report
Comments and Suggestions for Authors
The review explores the very interesting topic of the efficacy, safety and synergistic effect of vaccines in patients undergoing treatment with checkpoint inhibitors.
Although the authors aim to provide broad information on different kind of vaccines directed against different antigens, there are a few risks of bias that the authors should clarify. Due to the undoubted greater clinical experience of mRNA vaccines against SARS-CoV2 with respect to other clinical settings, in a few points of the review it is unclear whether the authors are referring to such specific vaccines or any mRNA vaccine. Likewise, when writing about cancer vaccines, it seems that the authors are mainly referring to mRNA vaccines, since many references on cancer vaccines plus ICI administration are missing.
In details:
SARS-CoV-2 mRNA Vaccination and ICI therapy
“Cytokine release syndrome (CRS) is a rare complication of both ICIs and mRNA vaccines. Yet, the combination of these agents results in a CRS like response observed alongside administration of SARS-CoV-2 mRNA vaccine with ICI therapy.” (pag.4)
The addition of the proper references for the above-mentioned statements will help clarify the extent of the CRS phenomenon for mRNA vaccines (which, in the reviewer’s opinion, is quite limited with only a few reports published in literature).
“Although both ICI-induced and mRNA vaccine-induced myocarditis may share a similar mechanism involving the overactivation of the immune response…”(pag.5)
Data reported on myocarditis cases observed post mRNA vaccination against SARS-CoV2 seem to suggest the involvement of the viral Spike protein. This should be considered when referring to possible common mechanisms for ICI and mRNA-induced myocarditis, since such kind of toxicity might not be a common feature for all mRNA vaccines.
CANCER DIRECTED VACCINES AND ICI-THERAPY
Therapeutic antigens as peptides have been presented alone or in combination with dendritic cells in combination with ICI therapy. A tumor antigen loaded dendritic cell vaccine in combination with ICI for stage IV melanoma was determined to be safe in a phase I trial, with four patients having stable disease 49. For recurrent HPV driven malignancies, the combination of ICI therapy and a long-peptide HPV-16 vaccine was used in a phase II trial and demonstrated a 33% overall response rate 50 (Pag. 8)
Due to the peculiar mechanisms of action of cell-based therapies, it is not correct to consider the administration of dendritic cells loaded with cancer peptides a “combination” of dendritic cells and peptides. The authors should modify that.
In general, the literature recalled for cancer vaccines different from mRNA vaccines is quite limited with respect to the plethora of data available on peptides, cell and adoptive cell transfer vaccination + ICI (see the reviews from Vafaei S et al, 2022 or Butterfiels et al, 2024). The author should implement this paragraph. Else, a selection limited to mRNA vaccines against cancer would be appropriate, but it should be declared in the title of the paragraph.
Author Response
Comment 1:
“Although the authors aim to provide broad information on different kind[s] of vaccines directed against different antigens, there are a few risks of bias that the authors should clarify. Due to the undoubted greater clinical experience of mRNA vaccines against SARS-CoV[-]2 with respect to other clinical settings, in a few points of the review it is unclear whether the authors are referring to such specific vaccines or any mRNA vaccine.”
Response:
Thank you for this insightful comment. We acknowledge the concern about potential bias in the discussion of mRNA vaccines, particularly regarding the frequent references to SARS-CoV-2 vaccines. To address this, we have revised the manuscript to clarify when we are referring to mRNA vaccines in general versus SARS-CoV-2-specific mRNA vaccines.
We have ensured that each instance now specifies the relevant context. For example, in the section discussing cytokine responses to vaccination, we explicitly state whether the observations pertain to SARS-CoV-2 mRNA vaccines or can be generalized to other mRNA platforms. This distinction ensures that our analysis is accurately framed and avoids any unintended overgeneralized bias toward SARS-CoV-2 vaccines alone.
These changes provide clear boundaries between the specific and general conclusions drawn from the literature, ensuring that our discussion is balanced and precise.
To address this, we updated key sections to specify the context for each reference to mRNA vaccines.
- On page 5, “In this review, we refer to mRNA vaccines both in the context of SARS-CoV-2 vaccines and more broadly to explore their potential interaction with ICI therapy beyond COVID-19. While the clinical experience with SARS-CoV-2 mRNA vaccines has been substantial, generalizing insights from these vaccines to other mRNA-based platforms warrants caution. Therefore, where appropriate, we specify whether the findings relate specifically to SARS-CoV-2 mRNA vaccines or reflect broader mRNA vaccine technology. This distinction ensures a balanced discussion and avoids bias in interpreting the safety, efficacy and immune responses associated with mRNA vaccines.”
- On page 6, “Due to the extensive data available for SARS-CoV-2 mRNA vaccines, many insights presented in this section reference these vaccines. However, emerging studies are beginning to evaluate other mRNA platforms for both infectious and cancer-directed vaccinations. While SARS-CoV-2 data provide a useful framework, the translation of these findings to other mRNA vaccines”
Comment 2:
“’Cytokine release syndrome (CRS) is a rare complication of both ICIs and mRNA vaccines.
Yet, the combination of these agents results in a CRS like response observed alongside
administration of SARS-CoV-2 mRNA vaccine with ICI therapy.’ The addition of the proper references for the above mentioned statement will help clarify the extent of the CRS phenomenon for mRNA vaccines.
Response:
Thank you for this insightful comment. We have revised the text to improve clarity and included the appropriate references to substantiate the statement. The revised sentence now reads:
“The interaction between mRNA vaccines and ICI therapy produces a notable cytokine response. Cytokine release syndrome (CRS) is a rare complication of ICIs.”
To clarify further, we referenced the World Health Organization pharmacovigilance database, which reports a low incidence of CRS among ICI-treated patients, primarily associated with PD-1 and PD-L1 inhibitors. Additionally, we highlight that the cytokine response observed in patients may be driven by components specific to the SARS-CoV-2 mRNA formulation. Studies suggest that both the spike protein encoded by the vaccine and the lipid nanoparticle delivery system might contribute to these heightened immune responses. To reflect this, we now include a reference to a review exploring these potential mechanisms of increased cytokine activity in SARS-CoV-2 mRNA vaccines.
This is clarified on page 6, “It is also important to observe that these increased cytokine responses related to the SARS-CoV-2 mRNA vaccines may be unique to this particular formulation. Both the spike protein encoded or the lipid nanoparticle may be the instigator of the enhanced cytokine response observed in patients 36.”
These revisions ensure that the phenomenon of CRS in the context of mRNA vaccines and ICIs is accurately described and supported by the relevant literature.
Comment 3:
“’Although both ICI-induced and mRNA vaccine-induced myocarditis may share a similar mechanism involving the overactivation of the immune response…’ Data reported on myocarditis cases observed post-mRNA vaccination against SARS-CoV-2 seem to suggest the involvement of the viral Spike protein. This should be considered when referring to possible common mechanisms for ICI and mRNA-induced myocarditis, since such kind of toxicity might not be a common feature for all mRNA vaccines.”
Response:
Thank you for this valuable observation. We have revised the manuscript to better reflect the nuances regarding myocarditis mechanisms associated with mRNA vaccines and ICIs. Specifically, the statement now clarifies that while both ICI-induced and SARS-CoV-2 mRNA vaccine-induced myocarditis may involve immune system overactivation, the mechanism driving myocarditis in SARS-CoV-2 mRNA vaccines appears to be more closely linked to the viral Spike protein.
Given that current data suggest the Spike protein may play a role in triggering myocarditis following SARS-CoV-2 vaccination, it is essential to acknowledge that not all mRNA vaccines encode the same antigens or use the same delivery platforms. Therefore, it cannot be assumed that myocarditis is a common feature of all mRNA vaccines. We have included this consideration to avoid overgeneralization and to ensure an accurate reflection of the available evidence.
This distinction is now integrated into the revised manuscript as follows: Page 7, “Additionally, although myocarditis has been observed with both ICI therapy and SARS-CoV-2 mRNA vaccination, the mechanism may be driven by the spike protein encoded by the SARS-CoV-2 vaccines. Therefore, myocarditis may not be a generalizable adverse effect across all mRNA vaccines. Careful monitoring and further evaluation of myocarditis risks with other mRNA vaccines are essential, particularly in ICI-treated patients."
Comment 4:
“When writing about cancer vaccines, it seems that the authors are mainly referring to mRNA vaccines, since many references on cancer vaccines plus ICI administration are missing. Additionally, due to the peculiar mechanisms of action of cell-based therapies, it is not correct to consider the administration of dendritic cells loaded with cancer peptides a ‘combination’ of dendritic cells and peptides. The authors should modify that. In general, the literature recalled for cancer vaccines other than mRNA vaccines is quite limited. The authors should implement this paragraph. Else, a selection limited to mRNA vaccines against cancer would be appropriate, but it should be declared in the title of the paragraph.”
Response:
Thank you for your thoughtful and constructive feedback. We have revised the section to address your concerns.
Clarifying the Scope of the Section: In response to your suggestion, we have expanded the discussion to include additional cancer vaccine platforms beyond mRNA vaccines, such as peptide-based and cell-based vaccines. We now clarify specific examples of long-peptide and dendritic cell vaccines used in combination with ICI therapy. The revised text can be found on page 10.
- Correcting Terminology for Dendritic Cell Vaccines: We acknowledge the need to accurately describe the role of dendritic cells in cancer vaccines. In the revised text, we have clarified that dendritic cells act as platforms for presenting tumor-associated antigens to activate T cells. We no longer describe this approach as a “combination” of dendritic cells and peptides, but rather as a targeted antigen-presentation strategy, in alignment with accepted terminology.
- Expanded Literature on Non-mRNA Cancer Vaccines: We have included additional references to recent reviews of combination strategies with ICI treatment. This ensures that the discussion reflects a broader range of vaccine platforms and highlights the diversity of cancer-directed combination strategies with ICI thearpy in development.
- Retaining the Current Section Title: Since the revised text now discusses multiple cancer vaccine platforms—including mRNA, peptide-based, and cell-based vaccines—we retained the original section title, "Cancer Directed Vaccines and ICI Therapy." However, we clarified within the text that mRNA vaccines are a focal point due to recent advancements but are not the sole focus of the section.
We believe these revisions address your concerns by broadening the discussion to reflect the diversity of cancer vaccine platforms, improving the terminology used, and ensuring that the section aligns with the latest literature on cancer vaccines combined with ICIs.
Reviewer 2 Report
Comments and Suggestions for Authors
The article titled "Immune Checkpoint Inhibitors and Vaccination: Assessing Safety, Efficacy, and Synergistic Potential" by Jacob et al. provides a thorough review of the safety, efficacy, and potential benefits of combining immune checkpoint inhibitors (ICIs) with infectious disease vaccines and therapeutic cancer vaccines. The review primarily focuses on influenza and mRNA SARS-CoV-2 vaccines, while also noting that data on other vaccines, such as pneumococcus, zoster, and HPV, remain limited. The article highlights the possible synergistic advantages of combining ICIs with vaccines, such as improved survival outcomes, and addresses the associated risk of immune-related adverse events (irAEs).
While the article is comprehensive and well-written, with strong potential for publication, some minor revisions could further enhance its quality. I would appreciate it if the authors could consider the following suggestions:
Ø There are minor typographical errors, such as the mention of “phase III date” instead of “phase III data,” which should be corrected. Additionally, acronyms like irAEs (immune-related adverse events) should be defined earlier in the text to improve clarity and readability.
Ø It may be beneficial for the authors to create separate sections for "Infectious Disease Vaccines" and "Therapeutic Cancer Vaccines." This structural change would improve the flow of the article and make it easier for readers to follow the different aspects of the discussion.
Ø The article mentions the synergistic potential of combining ICIs with mRNA-based therapeutic cancer vaccines but lacks sufficient detail on the current progress of clinical trials. Expanding this section with more information on ongoing or completed trials (such as those for melanoma and pancreatic cancer) would add greater depth to the review. Additionally, providing a critical analysis of the reasons behind the lack of phase III data would enhance the discussion.
Ø The supplementary table should be included in the main body of the article to provide the reader with direct access to important data without needing to refer to supplementary materials.
Comments on the Quality of English LanguageI think the article is well written and comprehensive.
Author Response
Comment 1
“While the article is comprehensive and well-written, with strong potential for publication, some minor revisions could further enhance its quality. I would appreciate it if the authors could consider the following suggestions: There are minor typographical errors, such as the mention of “phase III date” instead of “phase III data,” which should be corrected. Additionally, acronyms like irAEs (immune-related adverse events) should be defined earlier in the text to improve clarity and readability. “
Response:
Thank you for your positive feedback and helpful suggestions and apologize for the inclusion of these errors in the previous version of the manuscript. In response to your comment, we have carefully reviewed the manuscript to address the points you raised. All identified typographical errors, including the change from "phase III date" to "phase III data," have been corrected to ensure precision. Additionally, we have ensured that the acronym "irAEs" (immune-related adverse events) is clearly defined upon its first mention in the introduction, improving clarity and readability throughout the text. These adjustments were made to enhance the manuscript's polish and flow. We appreciate your thoughtful review and believe these changes strengthen the overall quality of the work.
Comment 2
It may be beneficial for the authors to create separate sections for ‘Infectious Disease Vaccines’ and ‘Therapeutic Cancer Vaccines.’ This structural change would improve the flow of the article and make it easier for readers to follow the different aspects of the discussion.
Response:
Thank you for this valuable suggestion. In response, we have adjusted the section titles to better reflect the distinct focus areas. The revised section titles are now “Infectious Disease Vaccinations and ICI Therapy” and “Cancer Directed Vaccines and ICI Therapy.” These changes highlight the distinction between the two types of vaccines, improving the clarity and flow of the manuscript.
Reviewer 3 Report
Comments and Suggestions for Authors
This review paper made up for the gaps between immune checkpoint inhibitors (ICI) and vaccines, especially the overall data on other infection disease vaccines are not enough. The author discussed the checkpoint inhibitors’ safety considerations, different types of vaccines in cancer ICI therapy. The construction of this review is very organized. I am wondering if the authors can add a cartoon to illustrate the current gaps of ICI therapy and vaccines either by the end of the review or at the beginning of the review.
Author Response
Comment 1:
“This review paper made up for the gaps between immune checkpoint inhibitors (ICI) and vaccines, especially the overall data on other infectious disease vaccines are not enough. The author discussed the checkpoint inhibitors’ safety considerations, different types of vaccines in cancer ICI therapy. The construction of this review is very organized. I am wondering if the authors can add a cartoon to illustrate the current gaps of ICI therapy and vaccines either by the end of the review or at the beginning of the review.”
Response:
Thank you for your thoughtful feedback and positive comments on our work. In response to your suggestion, we have created a visual illustration that highlights the current gaps and key outcomes related to the combination of immune checkpoint inhibitor (ICI) therapy with both infectious disease and cancer-directed vaccines. This figure (now included as NEW Figure 1) summarizes the available data on infectious disease vaccination and cancer-related vaccination outcomes when combined with ICI therapy.